# Establishing Aspergillus-Specific IgG Cut-Off Level for Chronic Pulmonary Aspergillosis Diagnosis: Multicenter Prospective Cohort Study

**DOI:** 10.3390/jof7060480

**Published:** 2021-06-12

**Authors:** Meng-Rui Lee, Hung-Ling Huang, Li-Ta Keng, Hsu-Liang Chang, Chau-Chyun Sheu, Pin-Kuei Fu, Jann-Yuan Wang, Inn-Wen Chong, Jin-Yuan Shih, Chong-Jen Yu

**Affiliations:** 1Department of Internal Medicine, National Taiwan University Hospital, Taipei 10002, Taiwan; leemr@ntu.edu.tw (M.-R.L.); ltkeng@gmail.com (L.-T.K.); jyshih@ntu.edu.tw (J.Y.-S.); jefferycjyu@ntu.edu.tw (C.-J.Y.); 2Department of Internal Medicine, National Taiwan University Hospital Hsin-Chu Branch, Hsin-Chu 30059, Taiwan; 3Department of Internal Medicine, Kaohsiung Municipal Ta-Tung Hospital, Kaohsiung 80145, Taiwan; 990325kmuh@gmail.com (H.-L.H.); hsuliac@gmail.com (H.-L.C.); 4Division of Pulmonary and Critical Care Medicine, Kaohsiung Medical University Hospital, Kaohsiung, Kaohsiung 80708, Taiwan; sheucc@gmail.com (C.-C.S.); chong@kmu.edu.tw (I.-W.C.); 5Department of Internal Medicine, Kaohsiung Medical University Hospital, Kaohsiung, Kaohsiung 80708, Taiwan; 6Graduate Institute of Medicine, College of Medicine, Kaohsiung Medical University, Kaohsiung 80708, Taiwan; 7Department of Critical Care Medicine, Taichung Veterans General Hospital, Taichung 40705, Taiwan; 8Ph.D. Program in Translational Medicine, National Chung Hsing University, Taichung 40254, Taiwan; 9College of Human Science and Social Innovation, Hungkuang University, Taichung 43302, Taiwan; 10Department of Computer Science, Tunghai University, Taichung 40704, Taiwan; 11Departments of Respiratory Therapy, Kaohsiung Medical University Hospital, Kaohsiung 80708, Taiwan; 12Department of Biological Science and Technology, National Yang Ming Chiao Tung University, Hsinchu 30010, Taiwan

**Keywords:** *Aspergillus fumigatus*, *Aspergillus* IgG, chronic pulmonary aspergillus, cut-off value, Taiwan

## Abstract

Objectives: *Aspergillus*-specific IgG (*Asp*-IgG) cut-off level in diagnosing chronic pulmonary aspergillosis (CPA) remains unknown. Methods: We prospectively recruited participants with clinical suspicion of CPA in three centers in Taiwan during 2019 June to 2020 August. Serum *Aspergillus fumigatus*-specific IgG (*Asp*-IgG) (Phadia, Uppsala, UPPS, Sweden) was examined. Optimal cut-off level was determined by Youden’s index and validated. Results: A total of 373 participants were recruited. In the derivation cohort (*n* = 262), *Asp*-IgG had an area under the receiver-operating-characteristic curve (AUC) of 0.832. The optimal cut-off level was 40.5 mgA/L. While applying this cut-off level to the validation cohort (*n* = 111), the sensitivity and specificity were 86.7% and 80.2%. Lowering the cut-off level from 40.5 to 27 mgA/L, the sensitivity was steady (30/36, 83.3% to 31/36, 86.1%) while specificity dropped from 81.9% (276/337) to 63.5% (214/337). Restricting CPA diagnosis to only chronic cavitary pulmonary aspergillosis (CCPA) and chronic fibrosing pulmonary aspergillosis (CFPA) yielded a cut-off level of 42.3 mgA/L in the derivation cohort with a sensitivity of 100% and specificity of 84.4% in the validation cohort. Conclusions: Serum *Asp*-IgG performs well for CPA diagnosis and provides a low false-positive rate when using a higher cut-off level (preferably around 40 mgA/L).

## 1. Introduction

Chronic pulmonary aspergillosis (CPA) is an important and emerging clinical infectious disease [1,2,3]. Owing to the growing population of cured TB and chronic lung disease, the disease burden of CPA is in the increase and poses a public health threat [4,5,6,7]. As there is improved awareness of this disease entity among physicians, more patients were being suspected and diagnosed for CPA. Diagnosis of CPA, however, can be difficult due to the low sensitivity of fungus culture for *Aspergillus,* and invasive diagnostic procedures were often intolerable and risky among vulnerable patients due to the severity of underlying structural lung diseases [1]. Blood serology test such as galactomannan (GM) antigen test is widely used in diagnosis of invasive pulmonary aspergillosis (IPA) but exerts low sensitivity in the diagnosis of CPA [1,8]. Antibody response, namely *Aspergillus*-specific IgG (*Asp*-IgG), is now considered a key diagnostic tool for CPA, with the advantage of being less invasive and having better sensitivity and specificity [1,9]. 

The presence of elevated *Asp*-IgG, however, can represent either active or past infection [10]. Its optimal cut-off level under different geographic area also remained indeterminate [11,12]. In our previous study, we found that the baseline *Asp*-IgG level in an Asian population with intermediate TB burden can be higher than expected [2]. This raises concerns whether *Asp*-IgG could be used for diagnosing CPA in this setting and an optimal cut-off level is also not readily available. Furthermore, previous studies determining the cut-off level of *Asp*-IgG were mainly conducted with retrospective design, using cases from CPA databases and healthy control [13,14,15,16]. Prospective studies in a real-world clinical setting were scant and multicenter recruitment as well as validation was also lacking. 

This study, therefore, aimed to investigate the clinical application of serum *Asp*-IgG on diagnosing CPA in Taiwan. By prospectively recruiting participants who were clinical suspects of CPA, we intended to find and validate an optimal cut-off level for CPA in a multicenter setting in different geographic areas in Taiwan.

## 2. Materials and Methods

### 2.1. Ethics Statement

The Institutional Review Boards of participating hospitals approved the study prior to study initiation and consent was obtained from all participants. 

### 2.2. Study Participants and Setting

This was a multicenter prospective study conducting in three hospitals in northern (National Taiwan University Hospital Hsin-Chu branch [NTUH-HC]), middle (Taichung Veterans General Hospital [TCVGH]) and southern (Kaohsiung Medical University Hospital Kaohsiung Municipal Ta-Tung Hospital [KMUH-MT]) Taiwan between June 2019 and August 2020. We recruited participants whose CPA was suspected clinically in the respiratory outpatient department of participating hospitals. Inclusion criteria were presence of chronic airway symptoms (cough, hemoptysis, dyspnea, sputum production) for at least 3 months and radiographic findings (cavities, pleural thickening, fibrotic change, nodules, consolidation) from which CPA could not be readily excluded. Those judged to be not having CPA will be followed for at least additional 3 months for symptoms and radiographic monitoring to readily exclude CPA. 

### 2.3. Derivation and Validation Dataset

Participants recruited and data collected in the NTUH-HC and TCVGH were included in the derivation dataset while participants and data from the KMUH-MT were included in the validation dataset. 

### 2.4. Measurement of Serum Asp-IgG Level

Measurement of *Asp*-IgG was performed by using automated ImmunoCAP systems (Phadia; Uppsala, UPPS, Sweden) [14]. We examined *A. fumigatus*-specific IgG for each serum. 

### 2.5. Definition and Data Collection

The diagnosis of CPA in this study was made according to the guidelines established by the European Society for Clinical Microbiology and Infectious Diseases (ESCMID)/European Respiratory Society (ERS) and Infectious Disease Society of America (IDSA) [1,9]. Briefly, the diagnosis of CPA requires at least three months of clinical course with clinical symptoms, persistent or progressive image findings consistent with CPA, microbiologic or histologic evidence of *Aspergillus* infection and exclusion of alternative diagnoses (such as lung cancer, TB and nontuberculous mycobacteria (NTM)) [1,9]. *Asp*-IgG was not taken into consideration for diagnosis first because this study was intended to establish the cut-off level of *Asp*-IgG in Taiwan. 

The types of CPA were differentiated into chronic cavitary pulmonary aspergillosis (CCPA), chronic fibrosing pulmonary aspergillosis (CFPA), simple aspergilloma (SA), subacute invasive aspergillosis (SAIA) and *Aspergillus* nodule according to previous guideline [1]. The main image findings suggestive for CCPA were one or multiple cavities with or without fungal ball, for CFPA were extensive fibrosis with fibrotic destruction of at least two lobes of lung complicating CCPA, for SA were single pulmonary cavity containing a fungal ball, for SAIA were progressive consolidation and for *Aspergillus* nodule were one or more nodules with or without cavitation [1]. For interpreting radiographic findings, we reviewed and recorded chest computed tomography (CT) findings (*n* = 305, 81.8%), and if chest CT was not available, we reviewed chest X-rays (*n* = 68, 18.2%). 

The final establishment of CPA and non-CPA diagnosis was made by two chest specialists, who were blinded to the *Asp*-IgG results, from the other two hospitals other than the one to which the patient was admitted. The radiographic findings were also interpreted by the same two chest specialists, and the patients’ radiographic reports issued by radiologists were also reviewed. The two chest specialists then assessed patients’ clinical data, images and clinical course to make a diagnosis of CPA or non-CPA. While the diagnosis of CPA may not be clear-cut cross-sectionally, we also followed our CPA and non-CPA patients’ therapeutic response longitudinally. For instance, for a patient who was considered to fulfill the criteria of CPA at initial, clinical response (either symptomatic or radiographic) after 3 months of antifungal agents was also evaluated to further ascertain CPA diagnosis. For a patient who was considered not to meet the CPA diagnostic criteria at initial, their clinical response to treatment aimed at presumed diagnosis (e.g., antibacterial agents for bacterial lung abscess, anti-mycobacterial agents for TB or NTM) was also assessed to further ascertain the diagnosis of non-CPA. The abovementioned components were also integrated in the final diagnosis of CPA and non-CPA. In case of discrepancy, discussion was held between the two chest specialists. Among rare cases when consensus still could not be achieved, a third chest specialist was included to make the final decision.

### 2.6. Statistical Analysis

Inter-group differences were analyzed using independent-sampled *t* test or one-way ANOVA (analysis of variance) for continuous variables with normal distribution and homogeneity of variance. For other continuous variables we used non-parametric Mann–Whitney *U* test or Kruskal–Wallis test. Chi-square test was used for categorical variables. The proposed cut-off level of *Asp*-IgG was determined by the Youden’s index. Area under the receiver-operating-characteristics curve (AUC) was used to describe and compare the performance of *Asp*-IgG as diagnostic tool. Sensitivity and specificity were used for assessment of *Asp*-IgG. Confidence intervals were constructed using exact binomial confidence intervals. Sensitivity analysis was performed by including only CCPA and CFPA as CPA while CCPA and CFPA were the two most common and aggressive forms of CPA. All data analyses were performed using SAS v. 9.4 (SAS Institute Inc, Cary, NC, USA).

## 3. Results

### 3.1. Patients Recruitment 

The patient recruitment process is illustrated in Figure 1. Briefly, a total of 373 participants were included. The case numbers of derivation and validation cohort were 262 and 111, respectively. A total of 199 participants were not screened because the primary care physicians may not be aware of the possibility of CPA. These patients were therefore not screened, and they did not revisit the outpatient clinic thereafter or their symptoms/imaging lesions subsided thereafter.

### 3.2. Clinical Characteristics of Participants in the Derivation Cohort

The clinical characteristics of CPA participants in the derivation and validation cohort were described in Table 1. 

In the derivation cohort, the median age was 63-year-old. The most common underlying condition was diabetes mellitus (DM) (*n* = 53, 20.3%), followed by malignancy (*n* = 35, 13.4%). The most common underlying lung condition was old TB (*n* = 107, 40.8%), followed by bronchiectasis (*n* = 102, 38.9%). CPA patients had lower body-mass index (BMI) and were more likely to have positive respiratory tract Aspergillus culture and have cavitation and pleural thickening on chest images. Among the 21 CPA patients in the derivation cohort, 14 were CCPA, 3 were CFPA, 1 was aspergilloma, 1 was Aspergillus nodule and 2 were SAIA. 

### 3.3. Optimal Cut-Off Level and Diagnostic Performance of Asp-IgG in the Derivation Cohort

The *Asp*-IgG titer was higher in the CPA patients than non-CPA patients in the derivation (73.2 ± 24.2 vs. 26.7 ± 26.3 mgA/L, *p* < 0.001; Table 1 and Figure 2A). The AUC of *Asp*-IgG in the derivation cohort was AUC of 0.838 (95% CI, 0.781–0.875; Figure 2B). The best cut-off point determined by best Youden’s index was 40.5 mgA/L, which had 81.0% (95% CI, 58.1–98.3%) sensitivity and 82.6% (95% CI, 77.2–87.1%) specificity in the derivation cohort. 

### 3.4. Clinical Characteristics of Participants in the Validation Cohort

In the validation cohort, the median age was 67. The most common underlying condition was also DM (*n* = 25, 22.5%), followed by malignancy (*n* = 18, 16.2%). The most common underlying lung condition was bronchiectasis (*n* = 60, 54.1%), followed by old TB (*n* = 36, 32.4%). CPA patients were more likely to have prior TB history, positive respiratory tract *Aspergillus* culture, pulmonary cavitation and bronchiectasis. 

### 3.5. Supplementary and Detailed Information of the Entire Cohort

Among the 15 CPA patients in the validation cohort, 7 were CCPA, 4 were CFPA, 1 was aspergilloma, 2 were *Aspergillus* nodule and 1 was SAIA.

For the entire cohort, none of the non-CPA patients developed CPA or invasive pulmonary aspergillosis during follow-up. Moreover, for all the non-CPA patients with Aspergillus culture-positivity, *Aspergillus* colonization was considered due to good response to treatment aimed at alternative diagnosis.

For the three cases of *Aspergillus* nodule in our cohort, biopsy was taken to rule out lung cancer. Among the 4 patients with malignancy in our CPA patients, two were lung cancer, one was nasopharyngeal cancer, and one was ureter cancer. All four patients received biopsy to exclude lung cancer and demonstrated therapeutic response after anti-fungal therapy.

### 3.6. Validation of Derived Asp-IgG Cut-Off Level in the Validation Cohort

The *Asp*-IgG titer was higher in the CPA patients in validation cohort (97.7 ± 59.2 vs. 26.2 ± 21.0 mgA/L, *p* < 0.001; Table 1 and Figure 2C). AUC of *Asp*-IgG in the validation cohort was 0.885 (95% CI, 0.808–0.936; Figure 2D) and using 40.5 mgA/L as cut-off level yielded sensitivity of 86.7% (95% CI, 59.5%–98.3%) and specificity of 80.2% (95% CI, 70.8–87.6%).

### 3.7. Asp-IgG Cut-Off Level Performance in the Entire Cohort

For the entire CPA cohort, the sensitivity and specificity of *Asp*-IgG at cut-off level of 40.5 mgA/L were 83.3% (95% CI, 67.2–93.6%) and 81.9% (95% CI, 77.4–85.9%). At 27 mgA/L, the sensitivity and specificity were 86.1% (95% CI, 70.5–95.3%) and 63.5% (95% CI, 58.1–68.7%).

### 3.8. Subgroup Analysis of Proposed Asp-IgG Cut-Off Level in CPA

The subgroup analysis of proposed *Asp*-IgG cut-off level in CPA was illustrated in Figure 3. The performance of proposed cut-off level (40.5 mgA/L) was satisfactory across different subgroups. In subgroups among which CPA may be of particular concern such as old TB patients, the sensitivity and specificity were 90.9% (95% CI, 70.8–98.9%) and 81.8% (95% CI, 73.8–88.2%). In patients with pulmonary cavitation, the sensitivity and specificity were 87.5% (95% CI, 71.0–96.5%) and 75.4% (95% CI, 63.1–85.2%). In patients with bronchiectasis, the sensitivity was 84.2% (95% CI: 60.4—96.7%), and specificity was 81.1% (95% CI: 73.7—87.2%). 

Separating the entire cohort into male and female patients, the optimal cut-off level and AUC would be 42.3 mgA/L and 0.918 among male patients, and this would lead to a sensitivity of 88.9% (95% CI: 65.3—98.6%) and specificity of 87.5% (95% CI: 82.0—91.8%). In female patients, the optimal cut-off level and AUC would be 63.3 mgA/L and 0.790 and this would lead to a sensitivity of 61.1% (95% CI: 35.8—82.7%) and specificity of 91.7% (95% CI: 86.0—95.7%). Using 42.3 mgA/L in the male and 63.3 mgA/L in the female patients as cut-off level yielded a sensitivity of 75% (95% CI: 57.887.9%) and specificity of 89.3% (95% CI: 85.5–92.4%) in the entire cohort.

### 3.9. Optimal Cut-Off Level of Asp-IgG for CCPA and CFPA (Excluding Aspergilloma, Aspergillus Nodule and SAIA)

After excluding CPA cases with aspergilloma, *Aspergillus* nodule and SAIA, the AUC of *Asp*-IgG in the derivation cohort was 0.852 (95% CI, 0.722–0.981). The optimal cut-off level determined by Youden’s index was 42.3 mgA/L in the derivation cohort, with a sensitivity of 82.4% (95% CI: 56.6–96.2%) and specificity of 85.1% (95% CI: 79.9–89.3%). Applying this cut-off level to the validation cohort, the sensitivity was 100.0% (95% CI: 71.5–100%) and specificity was 84.4% (95% CI: 75.5–91.0%).

### 3.10. Aspergillus IgG Level in Patients with Different CPA Types

Asp-IgG level was highest among patients with CFPA (*n* = 7, 143 ± 48.1 mgA/L), followed by aspergilloma (*n* = 2, 88.8 ± 12.7 mgA/L) and CCPA (*n* = 21, 73.3 ± 39.6 mgA/L). Asp-IgG level for aspergillus nodule (*n* = 3) and SAIA (*n* = 3) was 37.3 ± 21.8 and 57.8 ± 77.3 mgA/L. 

### 3.11. Comparison of Asp-IgG Level Between Hospitals

CPA and non-CPA *Asp*-IgG level among three different hospitals were illustrated in Figure 4. Comparison between three hospitals, the non-CPA group *Asp*-IgG was not different between the three hospitals (NTUH-HC vs. TCVGH vs. KMUH-MT: 26.3 ± 26.1 vs. 27.8 ± 18.1 vs. 26.2 ± 21.0 mgA/L, *p* = 0.889). 

Comparing between CPA patients among three hospitals, we found that *Asp*-IgG in NTUH-HC was slightly lower than TCVGH and KMUH-MT (NTUH-HC vs. TCVGH, 57.1 ± 45.1 vs. 99.3 ± 29.9 mgA/L, *p* = 0.030; NTUH-HC vs. KMUH-MT, 57.1 ± 45.1 vs. 97.7 ± 59.2 mgA/L, *p* = 0.054).

### 3.12. Reasons for Exclusion of CPA among Those with Asp-IgG above 40.5 mgA/L in the Non-CPA Group 

Among the 61 non-CPA patients with *Asp*-IgG titer above 40.5 mgA/L, CPA was excluded according to the following reasons and rationale: responding to antibacterial agents (*n* = 32, 52.5%), active TB (*n* = 18, 29.5%), NTM lung disease (*n* = 4, 6.6%), lung cancer (*n* = 6, 9.8%) and allergic bronchopulmonary aspergillosis (*n* = 1, 1.6%).

## 4. Discussion

Our study is the single largest cohort investigating the optimal cut-off level of serum *Asp*-IgG for CPA diagnosis. We found that a cut-off level at 40.5 mgA/L provided good sensitivity and specificity in both derivation and validation cohort in Taiwan. The more chronic and extensive form of CPA including CFPA, CCPA and aspergilloma had higher *Asp*-IgG compared with other forms of CPA. While different cut-off points have been proposed, a higher than traditionally considered level may reduce false-positive results and maintain satisfying sensitivity. 

There were several commercial *Asp*-IgG kits available for diagnosing CPA [13,14,17]. Different optimal cut-off points of *Asp*-IgG have been proposed in different areas and for different commercial kits [13,14]. Among them, ImmunoCap *Asp*-IgG is one of the most commonly used kits with good performance. For ImmunoCap *Asp*-IgG, different cut-off points have been proposed, ranging from 20 to 50 mgA/L [13,14,15,16]. Table 2 summarizes previous reports and the performance of their proposed cut-off points in original and current study. When lowering the cut-off points to around 20 to 30 mgA/L, there was only minimal improvement in sensitivity, but the specificity dropped significantly. This finding suggested that using a lower cut-off level may lead to significant false-positive results, especially in the area with high background seroprevalence of *Aspergillus* IgG [2]. In our previous study, we found that there is a high baseline seroprevalence of *Asp*-IgG in Taiwanese population [2]. According to our previous report, 22% and 22.9% of middle-aged healthy control and old TB patients had baseline Asp-IgG above 40 mgA/L in Taiwan [2]. Comparing with the study conducted by Page et al., none of control group (from Uganda) had Asp-IgG level above 40 mgA/L [14]. Moreover, in the study conducted by Seghal et al. in India, only 1.7% of the control group had Asp-IgG above 27.3 mgA/L [16]. This current study further emphasized the importance of avoiding over-diagnosis of CPA with a lower *Asp*-IgG cut-off point in a high background titer setting.

In our study, the *Asp*-IgG of CPA in NTU-HC (located in northern Taiwan, mean ± sd, 57.1 ± 45.1 mgA/L) was lower than TCVGH (located in middle Taiwan, mean ± sd, 99.3 ± 29.9 mgA/L) and KMUH-MT (located in southern Taiwan, mean ± sd, 97.7 ± 59.2 mgA/L). While we used the same recruitment criteria, this finding may suggest that *Asp*-IgG may also vary according to geographic characteristics. Indeed, the climate in middle and southern Taiwan may be warmer and more humid than norther Taiwan, which may facilitate fungus growth and lead to higher *Asp*-IgG titer [2].

Furthermore, it may be interesting to investigate the performance of our proposed cut-off point among population other than Taiwanese. Applying our 40.5 mgA/L to the 241 British CPA patients and 100 Uganda blood donors in Page et al. study in 2016 would yield a sensitivity of 88% and a specificity of 100% [14]. When applied to 114 Belgian controls in Page et al. study in 2018, 40.5 mgA/L as cut-off point still had a specificity of 90% [13]. The high sensitivity and specificity by our cut-off point suggest that 40.5 mgA/L cut-off level may also be applicable in other countries. 

Notably, optimal cut-off level may vary with regards to the selection of diseases and non-diseased patients (control) [18,19]. Previous studies have shown that sensitivity of *Asp*-IgG was lower in simple aspergilloma compared with CCPA. Furthermore, SAIA is a more acute form of CPA [18,19]. Including these forms of CPA in our study may therefore lead to lower sensitivity. On the other hand, using bronchiectasis or COPD patients with *Aspergillus* colonization as control, for instance, will also yield different specificity results compared with our non-diseased patients. This study, however, aimed to investigate the cut-off level of *Asp*-IgG in real-world practice. While different physicians may hold discrepant criteria and thresholds for suspecting CPA and ordering *Asp*-IgG tests, we believe that our selected control may be more akin to the clinical setting.

We found that *Asp*-IgG was higher in CCPA, CFPA and aspergilloma compared with SAIA and *Aspergillus* nodule. Interestingly, the degree of *Asp*-IgG elevation seemed to be associated with the chronicity and disease extent of CPA. Having the highest *Asp*-IgG level among all CPA entities, CFPA is the most severe and late form of CPA with multi-lobar involvement and extensive destruction [1]. CFPA patients may represent the most prolonged and pronounced chronic infection of *Aspergillus* among CPA patients. CCPA is the most common form of CPA and was considered to develop into CFPA if left untreated. The formation of aspergilloma may also take several years [1,20]. On the other hand, SAIA develops within a shorter period, and *Aspergillus* nodule, often with limited disease extent, is an unusual presentation of CPA requiring tissue biopsy for definite diagnosis [1]. 

While CCPA and CFPA may be the more aggressive and typical forms of CPAs, we have performed sensitivity analysis by including only CCPA and CFPA as CPA diagnosis. The yielded optimal cut-off level of 42.3 mgA/L by this sensitivity analysis was close to the 40.5 mgA/L from the primary analysis. This sensitivity analysis was in line with the design of and makes our results more comparable with previous studies [18,19]. Despite the fact that CCPA and CFPA may have higher *Asp*-IgG than other forms of CPA, the optimal cut-off level remained similar. 

In a recent meta-analysis comparing diagnostic accuracy of 5 *Aspergillus*-specific antibodies (precipitin, IgA, IgM, IgG and IgG + IgM) for CPA, *Asp*-IgG was highlighted as the preferred test over other antibody tests in screening for CPA [21]. With sensitivity of *Asp*-IgG fixed at 0.9, the specificity was 0.9 (95% credible intervals: 0.86–0.95) for diagnosing CPA. This study, however, also highlighted the limitations of high risk of bias due to case-control design in most studies and different thresholds used in different studies [21]. The findings of our study, therefore, can add to current understanding of *Asp*-IgG in CPA.

Our study also has limitations. First, not all commercial kits were performed in our study. We, however, consider that while most commercial kits showed comparable performance, the concept that heightening cut-off level may reduce false positive results in CPA diagnosis is worth investigating. Second, this is not a multination study, and our findings may need further validation in other countries. Though we have surveyed published studies and compared between different datasets with promising results, the proposed cut-off level in our study might still not be applicable in populations of different ancestry or living in other latitudes. Third, unlike some diseases with gold standard for diagnosis (e.g., lung cancer with pathology), CPA is a disease which requires a combination of clinical, radiographic and mycologic criteria and *Asp*-IgG is also a part of diagnostic criteria [1]. The establishment and exclusion of diagnosis are therefore not always straightforward. This may cause ambiguity in separating CPA and non-CPA patients. Though we have tried to not judge recruited patients on cross-sectional data only, we acknowledge that issues of over-diagnosis and under-diagnosis may still exist. Noteworthy, we cannot exclude the possibility that a few CPA cases would remain undiagnosed, especially among those with high Asp-IgG titer in the non-CPA group. Last, the definite symptoms were not recorded in the case record form, and information on the initial symptoms was therefore incomplete.

In conclusion, we found that a higher serum *Aspergillus*-specific IgG cut-off level (preferably around 40 mgA/L) may help reduce false-positive results and maintain good sensitivity. *Aspergillus*-specific IgG level may also correlate with disease duration and severity among CPA patients.

## Figures and Tables

**Figure 1 jof-07-00480-f001:**
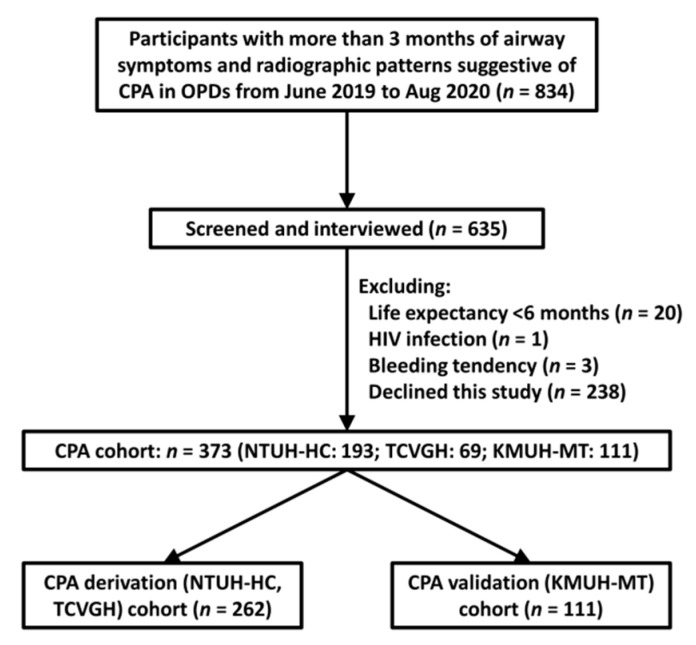
Patient recruitment process.

**Figure 2 jof-07-00480-f002:**
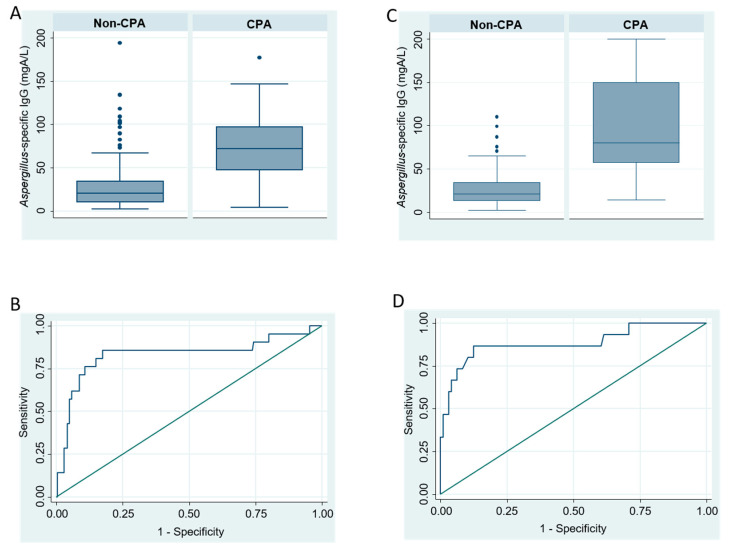
Boxplot and receiver operator characteristics (ROC) of serum Aspergillus fumigatus-specific IgG in chronic pulmonary aspergillosis (CPA) cohort ((**A)** boxplot in CPA derivation cohort; (**B**) ROC in CPA derivation cohort; (**C**) boxplot in CPA validation cohort; (**D**) ROC in CPA validation cohort).

**Figure 3 jof-07-00480-f003:**
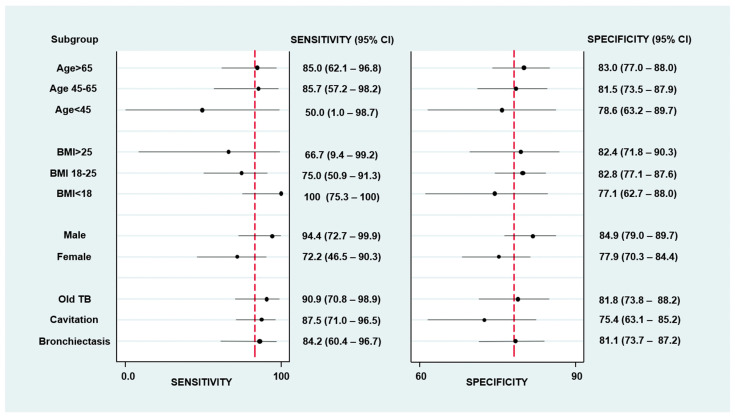
Subgroup analysis of serum Aspergillus fumigatus IgG performance in chronic pulmonary aspergillosis cohort.

**Figure 4 jof-07-00480-f004:**
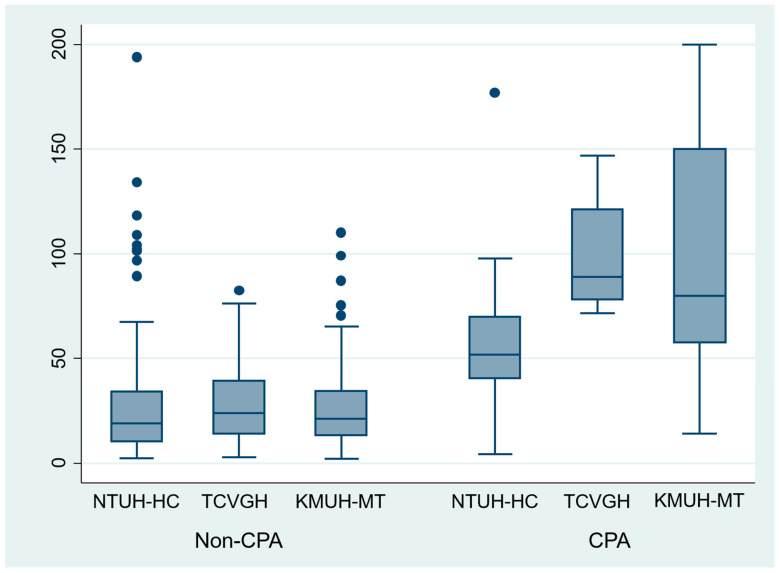
Boxplot of serum Aspergillus fumigatus IgG level among different groups in different hospitals.

**Table 1 jof-07-00480-t001:** Clinical characteristics of 373 participants.

	Derivation Cohort				Validation Cohort				
	All (*n* = 262)	CPA (*n* = 21)	Non-CPA (*n* = 241)	*P* * Value	All (*n* = 111)	CPA (*n* = 15)	Non-CPA (*n* = 96)	*P* ** Value	*P* *** Value
Age	62.1 ± 14.5	62.4 ± 11.6	62.0 ± 14.7	0.908	63.9 ± 14.8	66.3 ± 11.1	63.5 ± 15.4	0.501	0.266
<45	31 (11.8)	1 (4.8)	30 (12.5)	0.525	13 (11.7)	1 (6.7)	12 (12.5)	0.731	0.141
45–65	105 (40.1)	10 (47.6)	95 (39.4)		33 (29.7)	4 (26.7)	29 (30.2)		
>65	126 (48.1)	10 (47.6)	116 (48.1)		65 (58.6)	10 (66.7)	55 (57.3)		
Body mass index	22.2 ± 8.62	19.1 ± 2.9	22.5 ± 8.9	<0.001	22.2 ± 4.8	20.4 ± 3.4	22.5 ± 4.9	0.118	0.981
<18	43 (16.4)	9 (42.9)	34 (14.1)	<0.001	18 (16.2)	4 (26.7)	14 (14.6)	0.494	0.508
18–25	169 (64.5)	12 (57.1)	157 (65.2)		66 (59.5)	8 (53.3)	58 (60.4)		
>25	50 (19.1)	0	50 (20.8)		27 (24.3)	3 (20.0)	24 (25.0)		
Male	157 (59.9)	12 (57.1)	145 (60.2)	0.786	25 (22.5)	4 (26.7)	21 (21.9)	0.741	0.030
Underlying conditions									
DM	53 (20.3)	2 (9.5)	51 (21.2)	0.266	25 (22.5)	4 (26.7)	21 (21.9)	0.680	0.619
ESRD	1 (0.4)	0	1 (0.4)	0.767	1 (0.1)	0	1 (1.0)	1.000	0.507
Malignancy	35 (13.4)	2 (9.5)	33 (13.7)	0.590	18 (16.2)	2 (13.3)	16 (16.7)	0.745	0.470
Autoimmune disease	11 (4.2)	0	11 (4.6)	0.317	6 (5.4)	0	6 (6.3)	0.320	0.609
Immunosuppressant	2 (1.0)	2 (1.1)	0	0.702	12 (1.1)	2 (13.3)	10 (10.4)	0.735	<0.001
Old TB	107 (40.8)	12 (57.1)	95 (39.4)	0.113	36 (32.4)	10 (66.7)	26 (27.1)	0.002	0.127
Smoking	48 (18.3)	5 (23.8)	43 (17.8)	0.498	36 (32.4)	5 (33.3)	31 (32.3)	0.936	0.003
*Aspergillus* culture positivity	7 (2.7)	3 (14.3)	4 (1.7)	0.013	3 (2.7)	3 (20.0)	0	0.002	1.000
Serum GM (Index)	0.18 ± 0.37	0.41 ± 0.89	0.14 ± 0.14	0.280	0.17 ± 0.53	0.52 ± 1.46	0.12 ± 0.10	0.355	0.890
BAL GM (Index)	0.93 ± 1.81	2.81 ± 2.84	0.47 ± 1.10	0.394	0.19 ± 0.30	0.54 ± 0.70	0.15 ± 0.17	0.274	0.010
Chest image findings									
Cavitation	69 (23.4)	18 (85.7)	51 (21.2)	<0.001	28 (25.2)	14 (93.3)	14 (14.6)	<0.001	0.823
Nodule/Mass	149 (56.9)	11 (52.4)	138 (57.3)	0.665	69 (62.2)	12 (80.0)	57 (59.4)	0.126	0.486
Consolidation	141 (53.8)	14 (66.7)	127 (52.7)	0.218	42 (37.8)	8 (53.3)	34 (35.4)	0.183	0.005
Bronchiectasis	102 (38.9)	7 (33.3)	95 (39.4)	0.583	60 (54.1)	12 (80.0)	48 (50.0)	0.031	0.007
Pleural thickening	46 (17.6)	7 (33.3)	39 (16.4)	0.048	15 (13.5)	3 (20.0)	12 (12.5)	0.429	0.334
Ground-glass opacity	30 (11.5)	3 (14.3)	27 (11.2)	0.671	22 (19.8)	1 (6.7)	21 (21.9)	0.169	0.033
Pleural effusion	24 (9.2)	4 (19.1)	20 (8.3)	0.102	16 (14.4)	2 (13.3)	14 (14.6)	0.898	0.134
Fibrosis/Fibro-calcified lesion	65 (23.9)	3 (14.3)	62 (25.8)	0.241	31 (27.9)	5 (33.3)	26 (27.1)	0.616	0.542
*Aspergillus* IgG Titer (mgA/L)	30.4 ± 29.2	73.2 ± 24.3	26.7 ± 24.3	<0.001	35.9 ± 37.8	97.7 ± 59.2	26.2 ± 21.0	<0.001	0.173

BAL, bronchoalveolar lavage; CT, computed tomography; DM, diabetes mellitus; ESRD, end-stage renal disease; GM, galactomannan; TB, tuberculosis; * compare between derivation CPA and non-CPA; ** compare between validation CPA and non-CPA; *** compare between derivation and validation cohort. Data are either mean ± standard deviation or number (percentage).

**Table 2 jof-07-00480-t002:** Optimal cut-off values for Aspergillus-specific IgG proposed in literature and their performance in current study.

Studies	Case No.	Characteristics of Control	Cut-Off Level	Performance in Original Study			Country	Performance in Current Study	
				AUC	Sen (%)	Spe (%)		Sen (%)	Spe (%)
Page et al. 2016 [14]	CPA: 241 Control: 100	Healthy blood donors	20.0	0.996	96	98	CPA: British Control: Uganda	86 (71–95)	48 (42–53)
Huang et al. 2020 [15]	CPA: 35 Control: 50	Non-fungal respiratory diseases	21.7	0.934	86	92	Taiwan	86 (71– 95)	52 (46–57)
Sehgal et al. 2018 [18]	CPA (CCPA): 130 Control: 50	Treated pulmonary TB	27.3	0.976	95.6	100	India	86 (71–95)	64 (58–69)
Sehgal et al. 2019 [16]	Aspergilloma: 46 Control: 81	Treated pulmonary TB and non-CPA respiratory diseases	27.3	0.839	64	98	India	86 (71–95)	64 (58–69)
Page et al. 2018 [13]	CPA: 241 Control: 114	Healthy laboratory technicians	50.0	0.956	84	96	CPA: British Control: Belgium	75 (58–88)	90 (86– 93)
Current Study	CPA: 36 Control: 337	Non-CPA respiratory diseases	40.5	0.832	87 (validation cohort)	80 (validation cohort)	Taiwan	83 (67–94)	82 (77–86)

AUC, area under curve; CPA, chronic pulmonary aspergillosis; Sen, sensitivity; Spe, specificity.

## Data Availability

Data are available upon reasonable request.

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
