# Peer review of "Establishing Aspergillus-Specific IgG Cut-Off Level for Chronic Pulmonary Aspergillosis Diagnosis: Multicenter Prospective Cohort Study"

_jof, 2021, doi:10.3390/jof7060480_

Round 1
Reviewer 1 Report
Huang et al determine the optimal cut-off of a serological assay for the diagnosis of CPA. The strong point of this study is determining the cut off for Aspergillus IgG in CPA using an assay commonly used in practice, in a large population of patients with compatible findings: the control group in this study presented with similar clinical presentation as the patients eventually diagnosed with CPA, therefore this simulates real-life clinical practice.
There are some points that need addressing in the methodology and presentation of results:
Methods:
Line 76: the inclusion criteria need to be clarified: where were the patients recruited? was it a respiratory department? or general outpatient department? It is stated that patients had 'typically' symptoms for 3 months or radiological findings etc, but was this an inclusion criterion?
It is stated that those who did not have CPA were followed for 3 months to see if the diagnosis is eventually made. How many eventually had CPA among those?
A criterion was radiological findings compatible with CPA. More is needed on this: did one or more radiologists review the scans? What exactly were they looking for? It is mentioned they used published criteria, but it needs to be clarified. Eg was a consolidation enough to suggest CPA, or some criteria had to be present, eg cavity and pleural thickening? Did they review X-Rays or CT scans?
Results:
834 participants were eligible but only 635 screened. What happened to the remaining patients?
Why were patients with life expectance <6 months excluded? This might have excluded patients with severe COPD which is known to be a risk factor for CPA.
IT would be good to have something on the symptoms on presentation.
Table1: p value for BAL GM (0.394) is not significant despite clearly higher values in CPA, is this correct?
Half of the patients had a nodule or mass. How was the diagnosis of CPA made in these patients? In a patient with a lung mass, I would not rely eg on a sputum culture positive for aspergillus to rule out cancer. Did these patients have cancer ruled out?
Line 133: I would not consider diabetes to be an immunocompromising condition as such, perhaps call it underlying condition.
Line 134: malignancy is one of the underlying conditions in the cohort. Was it lung malignancy? Again, could it be they had lung cancer instead of CPA?
Table 2: this table is very useful, but it would even be better if you put a small comment on what the control groups were in each paper (eg healthy blood donors, patients with prior TB, patients presenting with similar symptoms etc). As you say in the discussion, the selection of the control group is most crucial in this type of analysis, and your study has the advantage of using a very clinically relevant patient population.
Line 204: why are the IgG levels so different between the 2 hospitals? 57 vs 99 seems like a big difference. Were the same criteria used in both hospitals? This has to be addressed in the discussion.
Author Response
Reviewer 1
Huang et al. determine the optimal cut-off of a serological assay for the diagnosis of CPA. The strong point of this study is determining the cut off for Aspergillus IgG in CPA using an assay commonly used in practice, in a large population of patients with compatible findings: the control group in this study presented with similar clinical presentation as the patients eventually diagnosed with CPA, therefore this simulates real-life clinical practice.
There are some points that need addressing in the methodology and presentation of results:
Ans: Thanks for the suggestions for the manuscript. We have addressed all the points and presented them below.
Methods:
Line 76: the inclusion criteria need to be clarified: where were the patients recruited? was it a respiratory department? or general outpatient department? It is stated that patients had 'typically' symptoms for 3 months or radiological findings etc, but was this an inclusion criterion?
Ans: Thanks for the excellent comments. We have revised the manuscript to make it more clear.
- Materials and Methods 2.2. Study Participants and Setting (Page 2): “We recruited participants who CPA was suspected clinically in the respiratory outpatient department of participating hospitals. Inclusion criteria were presence of chronic airway symptoms (cough, hemoptysis, dyspnea, sputum production) for at least 3 months and radiographic findings (cavities, pleural thickening, fibrotic change, nodules, consolidation) from which CPA could not be readily excluded.”
It is stated that those who did not have CPA were followed for 3 months to see if the diagnosis is eventually made. How many eventually had CPA among those?
Ans: Thanks for the excellent comments. None of the non-CPA patients developed CPA during follow-up. Together with the suggestions from Reviewer 2, We have added a sentence in the result section.
- Results 3.4. Clinical characteristics of participants in the validation cohort (Page 6): “For the entire cohort, none of the non-CPA patient developed CPA or invasive pulmonary aspergillosis during follow-up.”
A criterion was radiological findings compatible with CPA. More is needed on this: did one or more radiologists review the scans? What exactly were they looking for? It is mentioned they used published criteria, but it needs to be clarified. Eg was a consolidation enough to suggest CPA, or some criteria had to be present, eg cavity and pleural thickening? Did they review X-Rays or CT scans?
Ans: Thanks for the excellent comments. We have revised and added a paragraph in the method section to make it more clear. While the interpretation of the results was mainly done by chest specialists, we would also review the radiologic reports issued by radiologists.
- Materials and Methods 2.5. Definition and Data Collection (Page 3): “The main image findings suggestive for CCPA were one or multiple cavities with or without fungal ball, for CFPA were extensive fibrosis with fibrotic destruction of at least two lobes of lung complicating CCPA, for SA were single pulmonary cavity containing a fungal ball, for SAIA were progressive consolidation, for Aspergillus nodule were one or more nodules with or without cavitation [1]. For interpreting radiographic findings, we reviewed and recorded chest computed tomography (CT) findings (n=305, 81.8%) and if chest CT was not available, we reviewed chest X-rays (n=68, 18.2%). The radiographic findings were interpreted by at least two chest specialists and the patients’ radiographic reports issued by radiologists were also reviewed.”
Results:
834 participants were eligible but only 635 screened. What happened to the remaining patients?
Ans: Thanks for the comments. The 199 participants were not screened because for the following reasons. First, the primary care physicians may not be aware of the possibility of CPA and thus these patients were not screened and they did not revisit the outpatient clinic thereafter or their symptoms/imaging lesions subsided thereafter.
Why were patients with life expectance <6 months excluded? This might have excluded patients with severe COPD which is known to be a risk factor for CPA.
Ans: Thanks for the excellent comments. Life expectancy less than 6 months was not recruited because of moral issues. These participants may be too fragile to take the risk of even receiving blood-drawing. We acknowledge that this might excluded certain CPA patients but due to the relatively small number of patients excluded (n=20), the difference may be low.
IT would be good to have something on the symptoms on presentation.
Ans: Thanks for the excellent comments. We have added a paragraph describing this. Also, we may need to admit that in case report form, we only recorded the presence of symptoms but did not record the definite types of symptoms
- Materials and Methods 2.2. Study Participants and Setting (Page 2): “Inclusion criteria were presence of chronic airway symptoms (cough, hemoptysis, dyspnea, sputum production)”
Table1: p value for BAL GM (0.394) is not significant despite clearly higher values in CPA, is this correct?
Ans: Thanks for the comments. We have checked and this number is correct. The p value is not significant may be because that BAL GM was not routinely performed among participants (n = 88), resulting in a wide 95% confidence interval.
Half of the patients had a nodule or mass. How was the diagnosis of CPA made in these patients? In a patient with a lung mass, I would not rely eg on a sputum culture positive for aspergillus to rule out cancer. Did these patients have cancer ruled out?
Ans: Thanks for the excellent comments. While there were half patients with lung nodules or masses, the main lesions for CPA diagnosis were not lung nodules (for instance, cavitation with paracavitary fibrosis, pleural thickening). For the only case of Aspergillus nodule in our cohort, biopsy was taken to rule out lung cancer.
Line 133: I would not consider diabetes to be an immunocompromising condition as such, perhaps call it underlying condition.
Ans: Thanks for the excellent comments. We have revised accordingly.
- Results 3.2. Clinical characteristics of participants in the derivation cohort (Page 6): “The most common underlying condition was diabetes mellitus (DM) (n = 53, 20.3%)”
- Results 3.4. Clinical characteristics of participants in the validation cohort (Page 6): “The most common underlying condition was also DM (n = 25, 22.5%).”
Line 134: malignancy is one of the underlying conditions in the cohort. Was it lung malignancy? Again, could it be they had lung cancer instead of CPA?
Ans: Thanks for the excellent comments. Among the 4 patients with malignancy, two were lung cancer, one was nasopharyngeal cancer and one was ureter cancer. All four patients received biopsy to exclude lung cancer and demonstrated therapeutic response after anti-fungal therapy.
Table 2: this table is very useful, but it would even be better if you put a small comment on what the control groups were in each paper (eg healthy blood donors, patients with prior TB, patients presenting with similar symptoms etc). As you say in the discussion, the selection of the control group is most crucial in this type of analysis, and your study has the advantage of using a very clinically relevant patient population.
Ans: Thanks for the excellent suggestions. We have added a column describing the control characteristics. The revised column is added in Table 2.
Line 204: why are the IgG levels so different between the 2 hospitals? 57 vs 99 seems like a big difference. Were the same criteria used in both hospitals? This has to be addressed in the discussion.
Ans: Thanks for the excellent suggestions. Same criteria were used in both hospitals. We hypothesized that the difference may be due to geographic variations and we also added a paragraph in discussion section.
- Discussion 3rd Paragraph (Page 11): “In our study, the Asp-IgG in NTU-HC (located in northern Taiwan) was lower than TCVGH (located in middle Taiwan) and KMUH-MT (located in southern Taiwan). While we used the same recruitment criteria, this finding suggests that Asp-IgG may vary according to geographic characteristics. Indeed, the climate in middle and southern Taiwan may be warmer and more humid than norther Taiwan, which may facilitate fungus growth and lead to a higher Asp-IgG titer [2].”
- Discussion (Page 10): “In our study, the Asp-IgG of CPA in NTU-HC (located in northern Taiwan, mean ± sd, 57.1 ± 45.1 mgA/L) was lower than TCVGH (located in middle Taiwan, mean ± sd, 99.3 ± 29.9 mgA/L) and KMUH-MT (located in southern Taiwan, mean ± sd, 97.7 ± 59.2 mgA/L). While we used the same recruitment criteria, this finding may suggest that Asp-IgG may also vary according to geographic characteristics. Indeed, the climate in middle and southern Taiwan may be warmer and more humid than norther Taiwan, which may facilitate fungus growth and lead to higher Asp-IgG titer [2].”
Reviewer 2 Report
The authors describe an interesting study about the optimal cut-off value of Aspergillus-specific IgG levels in Taiwanese CPA patients to improve the difficult diagnosis of this patient cohort. The manuscript is very well written and also the limitations of the study are discussed in an appropriate way. Therefore, I just have to add some minor comments, which were in my opinion unclear or would improve the manuscript.
1.) Table 1: what was the disease of the Aspergillus culture positive non-CPA patients? Were these invasive aspergillosis patients or just false positive cultures?
2.) According to the data in Figure 3, it seems that the cut-off value works better for male patients. Therefore, I would like to see the performance of the test with gender related cut-off values, one for male and one for female patients. Maybe this would even improve the sensitivity and specificity of the test? In contrast to the BMI, which would be the other obvious difference in performance, the different genders are almost equally distributed, which would allow the analysis in this study.
3.) Line 228: Is there data available that the high Asp-IgG baseline in Taiwanese people is special compared to other countries?
Author Response
The authors describe an interesting study about the optimal cut-off value of Aspergillus-specific IgG levels in Taiwanese CPA patients to improve the difficult diagnosis of this patient cohort. The manuscript is very well written and also the limitations of the study are discussed in an appropriate way. Therefore, I just have to add some minor comments, which were in my opinion unclear or would improve the manuscript.
Ans: Thanks for the encouragement and suggestions for the manuscript. We have addressed all the points and described them below.
Table 1: what was the disease of the Aspergillus culture positive non-CPA patients? Were these invasive aspergillosis patients or just false positive cultures?
Ans: Thanks for the excellent comments. Aspergillus culture-positivity among non-CPA patients were false-positive results. We have added a paragraph in the manuscript to make it more clearly.
3.Results 3.4. Clinical characteristics of participants in the validation cohort (Page 6): “For the entire cohort, none of the non-CPA patient developed CPA or invasive pulmonary aspergillosis during follow-up. Also, for all the non-CPA patients with Aspergillus culture-positivity, Aspergillus colonization was considered due to good response to treatment aimed at alternative diagnosis.”
According to the data in Figure 3, it seems that the cut-off value works better for male patients. Therefore, I would like to see the performance of the test with gender related cut-off values, one for male and one for female patients. Maybe this would even improve the sensitivity and specificity of the test? In contrast to the BMI, which would be the other obvious difference in performance, the different genders are almost equally distributed, which would allow the analysis in this study.
Ans: Thanks for the excellent comments. We have analyzed accordingly and presented it in the result section.
3.Results 3.7. Subgroup analysis of proposed Asp-IgG cut-off level in CPA (Page 8): “Separating the entire cohort into male and female patients, the optimal cut-off level and AUC would be 42.3 mgA/L and 0.918 among male patients and this would lead to a sensitivity of 88.9% (95% CI: 65.3-98.6%) and specificity of 87.5% (95% CI: 82.0-91.8%). In female patients, the optimal cut-off level and AUC would be 63.3 mgA/L and 0.790 and this would lead to a sensitivity of 61.1% (95% CI: 35.8-82.7%) and specificity of 91.7% (95% CI: 86.0-95.7%). Using 42.3 mgA/L in the male and 63.3 mgA/L in the female patients as cut-off level yielded a sensitivity of 75% (95% CI: 57.8-87.9%) and specificity of 89.3% (95% CI: 85.5-92.4%) in the entire cohort.”
3.) Line 228: Is there data available that the high Asp-IgG baseline in Taiwanese people is special compared to other countries?
Ans: Thanks for the excellent comments. We have added a sentence providing data in the discussion section.
4.Discussion 2nd Paragraph (Page 11): “According to our previous report, 22.0% and 22.9% of middle-aged healthy control and old TB patients had baseline Asp-IgG above 40 mgA/L in Taiwan [2]. Comparing with the study conducted by Page et al. in Uganda, none of control group had Asp-IgG level above 40 mgA/L [14]. Also, in the study conducted by Seghal et al. in India, only 1.7% of the control group had Asp-IgG above 27.3 mgA/L [16].”
Reviewer 3 Report
The authors present a study to determine the optimal Aspergillus Ab level to diagnose chronic pulmonary aspergillosis (CPA). The study was conducted in Taiwan. The rationale is to initially study a derivation cohort, followed by a validation cohort. They find that patients with CPA have higher levels of Aspergillus Ab compared with non-CPA patients (recruited from those with chronic symptoms and compatible radiology).
However, the basic premise seems flawed. They refer to guidelines, though there is no mention of quantitative assessment of Ab levels in the diagnosis of this condition; any positive value is significant. The test they use is described, though no cut-off value for a positive result is given. There is a definition of CPA, but that can't include Ab since that is what they are studying. There is in fact no specific definition for what the authors are considering a case of CPA and non-CPA. The number they arrive at seems arbitrary, since it is unclear how they separated the 2 groups.
It would seem more relevant to describe how patients with higher Ab levels progressed compared to those with lower levels, or responded to therapy.
Author Response
The authors present a study to determine the optimal Aspergillus Ab level to diagnose chronic pulmonary aspergillosis (CPA). The study was conducted in Taiwan. The rationale is to initially study a derivation cohort, followed by a validation cohort. They find that patients with CPA have higher levels of Aspergillus Ab compared with non-CPA patients (recruited from those with chronic symptoms and compatible radiology).
However, the basic premise seems flawed. They refer to guidelines, though there is no mention of quantitative assessment of Ab levels in the diagnosis of this condition; any positive value is significant. The test they use is described, though no cut-off value for a positive result is given. There is a definition of CPA, but that can't include Ab since that is what they are studying. There is in fact no specific definition for what the authors are considering a case of CPA and non-CPA. The number they arrive at seems arbitrary, since it is unclear how they separated the 2 groups.
It would seem more relevant to describe how patients with higher Ab levels progressed compared to those with lower levels, or responded to therapy.
Ans: Thanks for the excellent comments. We agree that while doing a diagnostic kit verification study, it is more complicated to define CPA than to define other diseases which have a gold standard (such as biopsy for lung cancer). The diagnosis of CPA requires a constellation of clinical parameters and possibly more importantly, exclusion of other diseases and judicious follow-up and evaluation. We, however, have tried to overcome the important issues you addressed by careful follow-up and assessment of patients and their response to either antifungal or antibacterial agents. We have addressed these issues both in the method and limitation sections to make it more clearly.
2. Materials and Methods 2.5. Definition and Data Collection (Page 3): “Notably, we also followed our CPA and non-CPA patients’ response to antifungal or antibacterial agents (symptomatic and/or radiographic improvement) change and symptoms to further ascertain their diagnosis. For instance, a CPA patient would experience symptomatic improvement after antifungal agents and non-CPA patients would benefit greatly from antibacterial agents. The abovementioned components were then taken together to make a final diagnosis of CPA and non-CPA.”
4. Discussion Limitations, 8th Paragraph (Page 11): “Third, unlike some diseases with gold standard for diagnosis (eg, lung cancer with pathology), CPA is a disease which requires a combination of clinical, radiographic and mycologic criteria and Asp-IgG is also a part of diagnostic criteria [1]. The establishment and exclusion of diagnosis are therefore not always straightforward. This may cause ambiguity in separating CPA and non-CPA patients. We, however, have not only judged recruited patients on cross-sectional data but also followed their response to treatment and clinical course longitudinally. This would further separate our CPA and non-CPA more clearly.”
Round 2
Reviewer 1 Report
Most comments have been addressed. However I still have some concerns on how the CPA diagnosis was made. This needs to be explained a bit more and limitations acknowledged:
You clarified that the response to treatment (antifungals or antibacterials) was taken as a criterion in order to classify them as CPA or non-CPA. This is a very vague statement and should be explained further. For example, did the two chest specialists (who also reviewed the radiology) decide after 3 or 6 months of antifungals that the patient was better (and in what way better? radiology, or clinical, or microbiologically?) to decide if this was CPA? Also, as you say the IgG was not taken as a criterion. However, you have to clarify if the IgG was available to the investigators during this study. If it was done as part of routine care, then there clearly is the potential for a bias (if the investigator has seen the result of the IgG, then he/she will be biased inevitably, even if they do not intend to include the IgG in the thought process).
Minor comments:
Figure 1: Please add the details about what happened to the 199 patients who were eligible but not recruited.
As you recorded only the presence or absence of compatible symptoms (not the actual symptoms), this should be mentioned in the limitations section.
Thank you for the clarification on the aspergillus nodules. Your comments should also be included in the manuscript. The same for the comments on the type of cancer.
Author Response
Most comments have been addressed. However I still have some concerns on how the CPA diagnosis was made. This needs to be explained a bit more and limitations acknowledged:
You clarified that the response to treatment (antifungals or antibacterials) was taken as a criterion in order to classify them as CPA or non-CPA. This is a very vague statement and should be explained further. For example, did the two chest specialists (who also reviewed the radiology) decide after 3 or 6 months of antifungals that the patient was better (and in what way better? radiology, or clinical, or microbiologically?) to decide if this was CPA? Also, as you say the IgG was not taken as a criterion. However, you have to clarify if the IgG was available to the investigators during this study. If it was done as part of routine care, then there clearly is the potential for a bias (if the investigator has seen the result of the IgG, then he/she will be biased inevitably, even if they do not intend to include the IgG in the thought process).
Ans: Thanks for the excellent suggestions. We have revised accordingly to make the descriptions clearer as well as acknowledged the limitations of definite diagnosis of CPA together with the comments from reviewer 3.
Page 3. 2.5. Definition and Data Collection
“The final establishment of CPA and non-CPA diagnosis was made by two chest specialists, who were blinded to the Asp-IgG results, from the other two hospitals other than the one patient was admitted in. The radiographic findings were also interpreted by the same two chest specialists and the patients’ radiographic reports issued by radiologists were also reviewed. The two chest specialists then assessed patients’ clinical data, images and clinical course to make a diagnosis of CPA or non-CPA. While the diagnosis of CPA may not be clear-cut cross-sectionally, we also followed our CPA and non-CPA patients’ therapeutic response longitudinally. For instance, for a patient who was considered to fulfill the criteria of CPA at initial, clinical response (either symptomatic or radiographic) after 3 months of antifungal agents was also evaluated to further ascertain CPA diagnosis. For a patient who was considered not to meet the CPA diagnostic criteria at initial, their clinical response to treatment aimed at presumed diagnosis (eg, antibacterial agents for bacterial lung abscess, anti-mycobacterial agents for TB or NTM) was also assessed to further ascertain the diagnosis of non-CPA. The abovementioned components were also integrated in the final diagnosis of CPA and non-CPA. In case of discrepancy, discussion was held between the two chest specialists. Among rare cases when consensus still can not be achieved, a third chest specialist was included to make the final decision.”
Page 12. 4. Discussion
“Though we have tried to not judge recruited patients on cross-sectional data only, we acknowledge that issues of over-diagnosis and under-diagnosis may still exist.”
Minor comments:
Figure 1: Please add the details about what happened to the 199 patients who were eligible but not recruited.
Ans: Thanks for the comments. We have revised accordingly and added in the results section
Page 4. 3. Results 3.1. Patients recruitment
“199 participants were not screened because the primary care physicians may not be aware of the possibility of CPA. These patients were therefore not screened and they did not revisit the outpatient clinic thereafter or their symptoms/imaging lesions subsided thereafter.”
As you recorded only the presence or absence of compatible symptoms (not the actual symptoms), this should be mentioned in the limitations section.
Ans: Thanks for the comments. We have addressed this point in the limitation section.
Page 12. 4. Discussion
“Last, the definite symptoms were not recorded in the case record form and information on the initial symptoms was therefore incomplete.”
Thank you for the clarification on the aspergillus nodules. Your comments should also be included in the manuscript. The same for the comments on the type of cancer.
Ans: Thanks for the comments. We have added a paragraph in the results section.
Page 6, 7 3. Results 3.5. Supplementary and detailed information of the entire cohort
“3.5. Supplementary and detailed information of the entire cohort”
“For the three cases of Aspergillus nodule in our cohort, biopsy was taken to rule out lung cancer. Among the 4 patients with malignancy in our CPA patients, two were lung cancer, one was nasopharyngeal cancer and one was ureter cancer. All four patients received biopsy to exclude lung cancer and demonstrated therapeutic response after antifungal therapy.”
Reviewer 3 Report
Control groups in Taiwan have Ab levels comparable to CPA patients in a significant proportion of patients. The authors have not demonstrated how high Ab levels confirm the diagnosis, apart from any positive Ab level, in conjunction with clinical and radiographic findings.
Author Response
Control groups in Taiwan have Ab levels comparable to CPA patients in a significant proportion of patients. The authors have not demonstrated how high Ab levels confirm the diagnosis, apart from any positive Ab level, in conjunction with clinical and radiographic findings.
Ans: Thanks for the comments. We acknowledge the limitations of our study. We have added a paragraph in the result section to further elucidate how we exclude those non-CPA patients with high Asp-IgG titer. Also, we have added this important opinion in our limitations.
Page 10. 3. Results
“3.12. Reasons for exclusion of CPA among those with Asp-IgG above 40.5 mgA/L in the non-CPA group”
“Among the 61 non-CPA patients with Asp-IgG titer above 40.5 mgA/L, CPA was excluded according to the following reasons and rationale: responding to antibacterial agents (n=32, 52.5%), active TB (n=18, 29.5%), NTM lung disease (n=4, 6.6%), lung cancer (n=6, 9.8%), and allergic bronchopulmonary aspergillosis (n=1, 1.6%).”
Page 12. 4. Discussion
“Though we have tried to not judge recruited patients on cross-sectional data only, we acknowledge that issues of over-diagnosis and under-diagnosis may still exist. Noteworthy, we can not exclude the possibility that a few CPA cases would remain undiagnosed, especially among those with high Asp-IgG titer in the non-CPA group.”